# A systemic review and network meta-analysis of accuracy of intraocular lens power calculation formulas in primary angle-closure conditions

Wenhan Lu[1,2], Yu Hou[1,2], Hongfang Yang[1,2]*, Xinghuai Sun[1,2,3]*

**1** Department of Ophthalmology & Visual Science, Eye & ENT Hospital, Shanghai Medical College, Fudan University, Shanghai, China, **2** NHC Key Laboratory of Myopia, Chinese Academy of Medical Sciences and Shanghai Key Laboratory of Visual Impairment and Restoration (Fudan University), Shanghai, China, **3** State Key Laboratory of Medical Neurobiology and MOE Frontiers Center for Brain Science, Institutes of Brain Science, Fudan University, Shanghai, China

\* xhsun@shmu.edu.cn (XS); hongfang_yang@126.com (HY)

## Abstract

### Background

For primary angle-closure and angle-closure glaucoma, the fact that refractive error sometimes deviates from predictions after intraocular lens (IOL) implantation is familiar to cataract surgeons. Since controversy remains in the accuracy of IOL power calculation formulas, both traditional and network meta-analysis on formula accuracy were conducted in patients with primary angle-closure conditions.

### Methods

A comprehensive literature search was conducted through Aug 2022, focusing on studies on intraocular lens power calculation in primary angle-closure (PAC) and primary angle-closure glaucoma (PACG). A systemic review and network meta-analysis was performed. Quality of studies were assessed. Primary outcomes were the mean absolute errors (MAE) and the percentages of eyes with a prediction error within ±0.50 diopiters (D) or ±1.00 D (% ±0.50/1.00 D) by different formulas.

### Results

Six retrospective studies involving 419 eyes and 8 formulas (Barrett Universal II, Kane, SRK/T, Hoffer Q, Haigis, Holladay I, RBF 3.0 and LSF) were included. SRK/T was used as a reference as it had been investigated in all the studies included. Direct comparison showed that none of the involved formula outperformed or was defeated by SRK/T significantly in terms of either MAE or % ±0.50/1.00 D (all *P*>0.05). Network comparison and ranking possibilities disclosed BUII, Kane, RBF 3.0 with statistically insignificant advantage. No significant publication bias was detected by network funnel plot.

**Data Availability Statement:** All relevant data are within the manuscript and its Supporting Information files.

**Funding:** This research project was supported by Shanghai Municipal Health Commission (20214Y0073) and Clinical Research Plan of SHDC (SHDC2020CR6029). The authors were also supported by grants from National Key Research and Development Program of China (2020YFA0112700); the State Key Program of National Natural Science Foundation of China (82030027), and the subject of major projects of National Natural Science Foundation of China (81790641). The funders had no role in study design, data collection and analysis, decision to publish, or preparation of the manuscript.

**Competing interests:** The authors have declared that no competing interests exist.

## Conclusions

No absolute advantage was disclosed among the formulas involved in this study for PAC/PACG eyes. Further carefully designed studies are warranted to evaluate IOL calculation formulae in this target population.

## Trail registration

**Registration:** PROSEPRO ID: CRD42022326541.

## Introduction

Primary angle-closure disease (PACD), as the leading cause of irreversible blindness in East Asia, has been projected to affect 32 million adults aged between 40 and 80 years old by 2040 [1]. Given the growing popularity of physical examination and people's concerns about their ocular health in past decades, ophthalmologists are constantly moving towards early detection and treatment of PACD. Though laser peripheral iridotomy (LPI) prophylaxis is a widely performed measure in patients at risk of angle-closure, it has little effect in treating existent peripheral anterior synechia (PAS) at intermediate and advanced stages [2]. Nowadays, cataract surgery (or clear-lens extraction) has been a preferred option for eyes with PACD due to its multiple benefits. i.e., improving eyesight, lowering intraocular pressure (IOP), and widening the narrowed anterior chamber angle [3, 4], yet the clinical benefits might be hindered by the dissatisfactory post-operative visual acuity.

Compared with cataract surgery in normal population, a range of publications have suggested that similar procedures in PACD patients often resulted in inaccurate refractive outcomes [3, 5]. According to current studies, for simple cataract patients, the percentage of post-operative refractive error within ±0.50 D varied between 73% and 88%, while the percentage dropped to 50%~68% for PACD patients [5, 6], indicating extra inaccuracy in terms of IOL power prediction in PACD patients, which had drawn ophthalmologists' attention. Along with well-accepted precision of ocular biometry, accuracy of manufacturer IOL power quality control and the now-mature surgical techniques, the accuracy of IOL power calculation becomes the most essential factor for achieving the target refraction post-operatively. A dozen formulas for IOL power calculation have been established with diverse core principles on the basis of normal eyes since 1980s, yet the accuracy varies [7–9]. Theoretically, it could be well imagined that discrepancies in IOL prediction would occur when these formulas, especially regressive ones [10], were applied to special population with unusual ocular biometric parameters, e.g., significantly shallow anterior chamber (AC), thick lens and short axial length (AL) [11–13]. Therefore, the final issue boils down to choosing or even developing a proper IOL calculation formula for PACD eyes to achieve optimal postoperative visual acuity after cataract surgeries [14, 15].

Hence, we performed a systemic review together with network meta-analysis of the accuracy of IOL power calculation formulas in patients with primary angle-closure conditions, hoping to figure out which formula currently available provides the highest accuracy for this target cohort by comparing and ranking those commonly used ones on their performance.

## Methods

Prior to analysis, a detailed protocol was made in accordance with the standard systemic review guidelines outlined by the Cochrane Reviewers' Handbook. The PRISMA checklist was

reviewed, strictly following the items required for systemic review and network meta-analysis (S1 Checklist) [16, 17]. Our research adhered to the tenets of the Declaration of Helsinki, and was registered on PROSPERO (CRD42022326541).

## Literature search

Two independent investigators (W.L. and Y.H.) conducted the comprehensive literature search in the following databases for relevant English language literature: PubMed, Cochrane Data Base of Systematic Reviews, the Cochrane Central Register of Controlled Trials (CENTRAL) and Web of Science through Aug 2022. The search terms were ("angle closure" OR "angle-closure") AND ("cataract" OR "IOL" OR "intraocular lens"). The detailed search results of that search process are provided in S1 Table. In order to ensure that no relevant study would be omitted, two independent authors (W.L. and Y.H.) also checked the reference lists for relevant studies.

## Inclusion and exclusion criteria

Inclusion criteria for studies were: 1) randomized controlled trials, cross-sectional studies, retrospective or prospective observational studies; 2) eyes diagnosed with PACS (primary angle-closure suspect, defined as eyes in which at least 180˚ of the posterior pigmented trabecular meshwork was invisible on gonioscopy in the primary position of gaze without indentation but with neither increased IOP nor glaucomatous neuropathy), PAC (defined as eyes with evidence of PAS or increased IOP on top of gonioscopic criteria for PACS), or PACG (defined as those PAC who additionally demonstrated glaucomatous damage of the optic nerve) [18]; 3) eyes undergoing uncomplicated cataract surgery or clear lens extraction with one-piece IOL implantation; 4) comparison of at least two types of IOL power calculation formulas available and 5) pre- and postoperative data collected with no less than 1-month interval, providing at least one outcome variable (i.e., mean absolute error (MAE) or percentages of predicted refractive error within a certain range) or one that can be calculated.

Exclusion criteria for studies were: 1) eyes with prior intraocular surgeries other than laser peripheral iridotomy (LPI), given that cataract surgery in patients with prior trabeculectomy had significantly greater refractive surprise than those in the control groups [19, 20]; 2) eyes with other conditions ineligible for inclusion (e.g. severe pterygium, severe corneal or vitreous opacity, macular degeneration, and retinal detachment, etc. [21]); 3) preoperative ocular biometric measurements performed with instruments other than IOLMaster; 4) studies that did not provide targeted outcome data or that were computationally unavailable.

## Data extraction and quality assessment

Two authors (W.L. and Y.H.) independently extracted the following data from each study: first author, year of publication and journal, country where the study was conducted, study period, study design, original inclusion criteria and exclusion criteria, total number of eyes included, demographic and biometry data of patients, IOLs types for each patient, patients' prior surgical history, formulas being evaluated, time intervals between pre- and postoperative biometric evaluation, and both outcomes measurements and values. Disagreements were resolved by discussion until consensus was reached or by consulting a third author (H.Y.).

The quality of studies were assessed using a modified Quality Assessment of Diagnostic Accuracy Studies-2 (QUADAS-2) tool designed for diagnostic tests [22], and the level of evidence were evaluated as recommended by the Oxford Centre for Evidence-based Medicine. Risk of publication bias were measured using network funnel plot.

## Outcome measurements

Mean absolute error (MAE), % ±0.50 D and % ±1.00 D were used as major outcomes of the review. MAE was defined as mean of the absolute value of the differences between the actual and the predicted postoperative spherical equivalent (SE) of refraction using each formula. % ±0.50 D (or % ±1.00 D) was defined as the percentage of eyes with prediction error ranging from -0.50 D (or -1.00 D) to + 0.50 D (or +1.00 D) of the target refraction after IOL implantation, as previously reported [15]. If any outcome value was not reported, we calculated it from the raw data provided by the studies. If the standard deviation (SD) data could not be retrieved from the text, this study was not eligible for MAE pooling and were therefore excluded.

## Statistical analysis

For traditional meta-analysis, STATA v 14.0 (StataCorp LP) was applied to generate head-to-head comparisons between SRK/T formula and the others. MAE was analyzed by weighted mean difference (WMD), and % ±0.50 D and % ±1.00 D were evaluated by the estimated risk ratios (RRs) with 95% credible intervals (CIs). An WMD less than 0 and RR greater than 1 indicated that the alternative formula was more accurate than SRK/T. $I^2$ value, together with the $P$ value of Cochran's Q test (for MAE) or Mantel Haenszel Q test (for % ±0.50/1.00 D), were reported for estimating the statistical heterogeneity in case of the bias introduced by small study effect [23]. An $I^2$ value greater than 50% with the $P$ value less than 0.05 was considered substantial heterogeneity, where random-effect model analysis was performed. Otherwise the fixed effect model was used. The risk for publication bias was measured by network funnel plots, and a symmetrical funnel plot indicated the relatively low risk of bias.

For network meta-analysis, pairwise analysis was conducted based on direct and indirect evidence by R v 3.5.2 (RStudio, Inc.). WMD of MAE, RRs with 95% credible intervals (CIs) for % ±0.50 D and % ±1.00 D were pooled. If no evidence of inconsistency was detected, the consistency model was adopted to calculate the relative effects of all investigated formulas. The Markov Chain Monte Carlo (MCMC) methods was applied to obtain the pooled effect sizes, with 4 chains calculated. Each chain comprised 20 000 simulation iterations, 5000 adaptation iterations and a thinning interval of 1. Statistically significant findings were recognized when bars of CIs did not include the value of 0 for the mean difference (MD) or 1 for the RR. Different formulas were ranked according to their performances of accuracy, i.e., the calculated possibilities of each formula at a specific rank based on their MAE, % ±0.50 D and % ±1.00 D, and the results were summarized and illustrated by bar plots.

## Results

### Literature search

Initial literature search identified 2752 articles (Fig 1). After comprehensive screening of the articles, 6 eligible studies were included in this systemic review and network meta-analysis, involving 419 eyes and 8 IOL calculation formulas. The demographic and other main characteristics extracted were summarized in Tables 1 and 2.

All studies included were designed as retrospective observational studies and mostly with a female-predominate pattern. Patients, if necessary, underwent prior surgeries but no more than LPI. One-piece IOL were implanted and all the ocular biometries were performed by IOLMaster preoperatively, with a follow-up period of no less than 1 month. For formulas, the overall network of the original direct comparisons of the 8 methods involved was illustrated in Fig 2. SRK/T had been investigated in all the studies included, therefore showing strongest links with the other formulas (Fig 2).

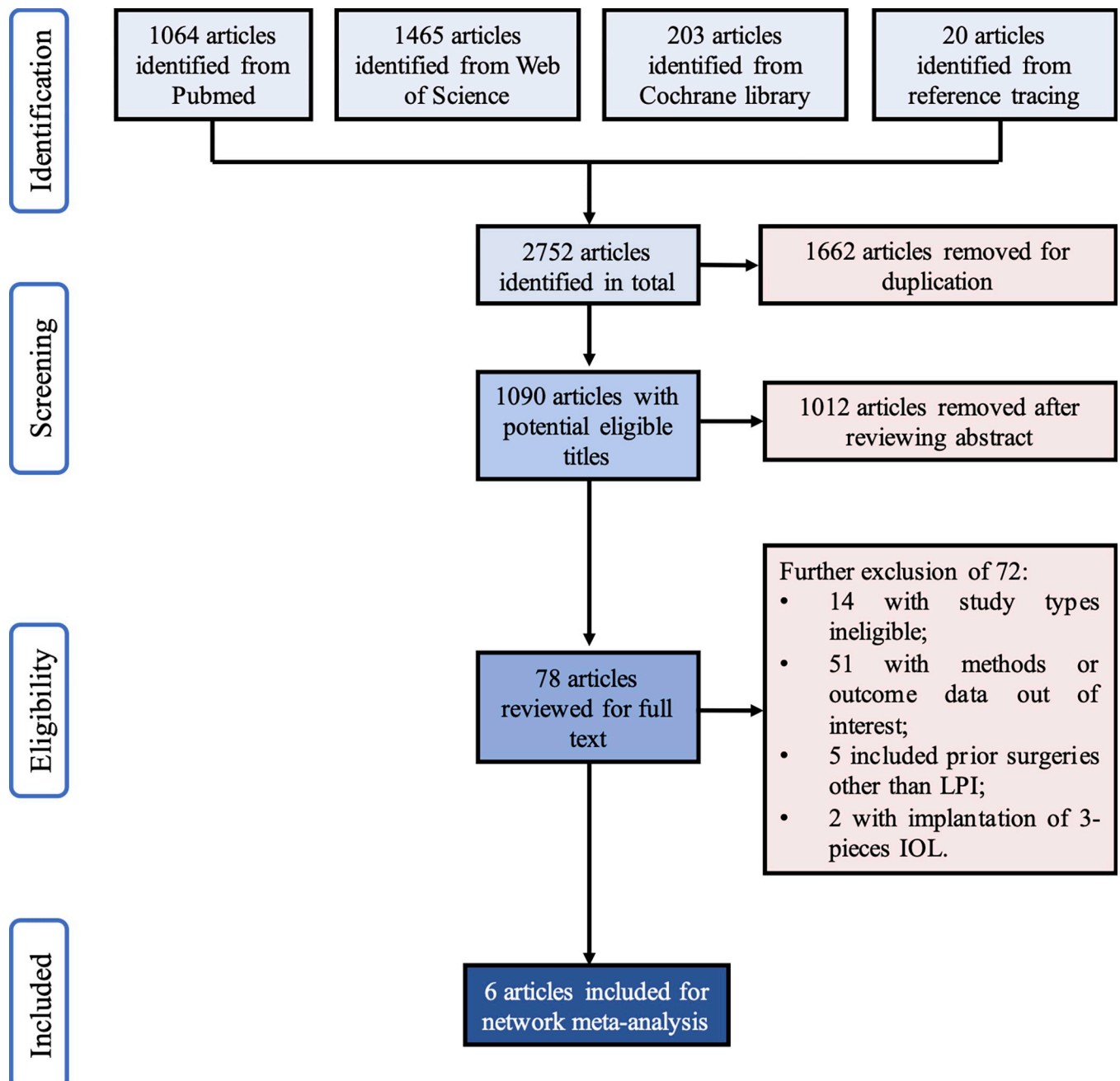

**Fig 1. Flowchart of study selection.** Shown is the process from database searching to the studies selected for this meta-analysis. After literature search and comprehensive selection, 6 articles were identified eligible for network meta-analysis. LPI: laser peripheral iridotomy; IOL: intraocular lens.

## Quality identification of included studies

The quality of each study evaluated by using the QUADAS-2 tool was plotted in Fig 3. Generally, all studies were of acceptable risk of bias. Further identification of risk of bias of included studies was summarized in S2 Table, with level of evidence recommended by the Oxford Centre for Evidence-based Medicine, indicating that all the studies included were qualified for this systemic review and meta-analysis.

**Table 1. Data extracted for basic characteristics of enrolled studies eligible for network meta-analysis.**

| Author/year | Design | Patients/eyes | Gender M/F | Mean age ± SD (years) | Preoperative Mean IOP ± SD (mmHg) | AL ± SD (mm) | ACD ± SD[a] (mm) | IOL (patient number) | Ocular biometry performed by | Constants optimization | LPI history | Examination intervals | Refractive outcome measured by | Available outcome measurements[b] | SRK/T | BUII | Kane | Hoffer Q | Haigis | Holladay I | RBF 3.0 | LSF |
|---|---|---|---|---|---|---|---|---|---|---|---|---|---|---|---|---|---|---|---|---|---|---|
| Joo [24], 2011 | Retrospective | 63/63 | N/A | 59.87 ± 10.55 | 16.32 ± 3.71 | 23.67 ± 1.03 | 2.28 ± 0.21 | SN60WF (63) | IOLMaster | ULIB | 100% | 3 months | AR | MAE; % ±0.50/1.00 D | ✓ | | | ✓ | ✓ | | | |
| Seo [7], 2016 | | 103/103 | 18/85 | 72.0 ± 7.0 | 15.93 ± 4.57 | 22.71 ± 0.69 | 2.23 ± 0.77 | SN60WF (103) | IOLMaster | Excel Query/ What IF function | N/A | > 1 month | SR | MAE; % ±0.50/1.00 D | ✓ | | | ✓ | ✓ | | | |
| Song [25], 2018 | | 50/50 | 8/42 | 69.4 ± 8.3 | 14.98 ± 3.54 | 22.64 ± 0.87 | 2.40[c] | Hoya iSert 250 (50) | IOLMaster | N/A | 34% | 1 month | SR | MAE; % ±0.50/1.00 D | ✓ | | | | ✓ | | | |
| Li [21], 2021 | | 111/111 | 33/78 | 64.21 ± 8.06 | N/A | 22.42 ± 0.87 | 2.31 ± 0.24 | 920H (29); 970C (82) | IOLMaster 700 | ULIB | 0 | 1–3 months | SR | MAE; % ±0.50/1.00 D | ✓ | ✓ | ✓ | ✓ | | | | |
| Lee [9], 2021 | | 43/43[d] | 9/34 | 70.1 ± 5.1 | N/A | 22.66 ± 0.64 | 2.42 ± 0.24 | SN60WF (25); ZCB00 (18) | IOLMaster | N/A | 0 | 1–3 months | AR | MAE | ✓ | | | ✓ | ✓ | ✓ | | |
| Hou [26], 2021 | | 49/49[e] | 15/34 | 64.4 ± 8.4 | N/A | 22.43 ± 0.88 | 2.22 ± 0.26 | ZCB00; MX60; SN60WF; Aspira-aA; 970C (N/A) | IOLMaster 700 | ULIB | 0 | > 1 month | SR | % ±0.50/1.00 D | ✓ | ✓ | ✓ | ✓ | ✓ | | ✓ | ✓ |

ACD: anterior chamber depth; AL: axial length; AR: auto-refractor; BUII: Barrett Universal II; F, female; IOP, intraocular pressure; LPI: laser peripheral iridotomy; LSF: Ladas Super Formula; MAE: mean absolute error; M, male; N/A, not available; RBF: Hill-Radial Basis Function; SD: standard deviation; SR: subjective manifest refraction; ULIB: the Users Group for Laser Interference Biometry.

[a] ACD measured from corneal epithelium to the anterior surface of the lens.

[b] The outcome measurements included only MAE and % ±0.50/1.00 D, which were to be analyzed.

[c] ACD was measured from corneal endothelium to the anterior surface of the lens, so was thereafter combined with corneal thickness for estimation of defined ACD.

[d] In this study, 70 PACD eyes were classified into PAS (+) (43 eyes) and PAS (−) (29 eyes). We only selected those with PAS (which stood for PAC/PACG eyes) and excluded PAS (−) as it included both partial PAC and PACS, the latter of which was out of our interest.

[e] In this study, 129 PAC/PACG eyes were enrolled and were divided into those with no prior surgeries (N = 49) and those with LPI or trabeculectomy (N = 94). We only selected those without prior surgeries for prevention of potential confusion brought by trabeculectomy.

**Table 2. Outcome values for quantitative analysis of enrolled studies.**

| Author/year | Patients/ eyes | Available outcome measurements | Outcome Values | | | | | | | |
|---|---|---|---|---|---|---|---|---|---|---|
| | | | SRK/T | BUII | Kane | Hoffer Q | Haigis | Holladay I | RBF 3.0 | LSF |
| Joo [24], 2011 | 63/63 | MAE (Mean [D] ± SD [D]) | 0.54 ± 0.47 | N/A | N/A | 0.53 ± 0.39 | 0.69 ± 0.54 | N/A | N/A | N/A |
| | | % ±0.50 D[a] | 35 (55.6%) | N/A | N/A | 43 (68.3%) | 31 (49.2%) | N/A | N/A | N/A |
| | | % ±1.00 D | 52 (82.5%) | N/A | N/A | 59 (93.7%) | 55 (87.3%) | N/A | N/A | N/A |
| Seo [7], 2016 | 103/103 | MAE (Mean [D] ± SD [D]) | 0.46 ± 0.34 | N/A | N/A | 0.44 ± 0.34 | 0.50 ± 0.37 | N/A | N/A | N/A |
| | | % ±0.50 D | 64 (62.1%) | N/A | N/A | 71 (68.9%) | 64 (62.1%) | N/A | N/A | N/A |
| | | % ±1.00 D | 90 (87.4%) | N/A | N/A | 94 (91.3%) | 88 (85.4%) | N/A | N/A | N/A |
| Song [25], 2018 | 50/50 | MAE (Mean [D] ± SD [D]) | 0.512 ± 0.328 | N/A | N/A | N/A | 0.593 ± 0.461 | N/A | N/A | N/A |
| | | % ±0.50 D | 22 (44%) | N/A | N/A | N/A | 25 (50%) | N/A | N/A | N/A |
| | | % ±1.00 D | 46 (92%) | N/A | N/A | N/A | 38 (76%) | N/A | N/A | N/A |
| Li [21], 2021 | 111/111 | MAE (Mean [D] ± SD [D]) | 0.69 ± 0.57 | 0.58 ± 0.57 | 0.66 ± 0.55 | 0.72 ± 0.59 | N/A | N/A | N/A | N/A |
| | | % ±0.50 D | 54 (48.6%) | 49 (44.1%) | 55 (49.5%) | 48 (43.2%) | N/A | N/A | N/A | N/A |
| | | % ±1.00 D | 79 (71.2%) | 82 (73.9%) | 82 (73.9%) | 81 (73.0%) | N/A | N/A | N/A | N/A |
| Lee [9], 2021 | 43/43 | MAE (Mean [D] ± SD [D]) | 0.65 ± 0.38 | N/A | N/A | 0.70 ± 0.46 | 0.61 ± 0.43 | 0.63 ± 0.41 | N/A | N/A |
| Hou [26], 2021 | 49/49 | % ±0.50 D | 33 (67.4%) | 31 (63.3%) | 35 (71.4%) | 29 (59.2%) | 32 (65.3%) | N/A | 35 (71.4%) | 29 (59.2%) |
| | | % ±1.00 D | 46 (93.9%) | 46 (93.9%) | 45 (91.8%) | 46 (93.9%) | 45 (91.8%) | N/A | 46 (93.9%) | 46 (93.9%) |

BUII: Barrett Universal II; LSF: Ladas Super Formula; MAE: mean absolute error; N/A, not available; RBF: Hill-Radial Basis Function; SD: standard deviation.

[a] % ± 0.50/1.00 D presented as number of eyes (percentage).

## Traditional meta-analysis

Traditional meta-analysis was conducted for direct comparison between the most commonly used SRK/T formula (which was evaluated by all the studies included) and the others. However, the single-study investigation of BUII, Kane and Holladay I formula for MAE comparison, and RBF 3.0, LSF formula for % ±0.50 D and % ±1.00 D comparison made them ineligible for pooled analysis. All pairs exhibited little heterogeneity as $I^2 < 50\%$ with $P$ value greater than 0.05 so that fixed model was applied. In terms of MAE, neither Haigis nor Hoffer Q showed significant superiority over SRK/T by direct comparison (Fig 4A, $P = 0.140$ between SRK/T and Haigis; $P = 0.990$ between SRK/T and Hoffer Q). With regard to % ±0.50 D and % ±1.00 D in comparison to SRK/T, Hoffer Q and Kane showed insignificantly higher percentage (Fig 4B, RR 1.03, $P = 0.434$ for % ±0.50 D by SRK/T-Hoffer Q; RR 1.05, $P = 0.250$ for % ±1.00 D by SRK/T-Hoffer Q; RR 1.03, $P = 0.731$ for % ±0.50 D by SRK/T-Kane and RR 1.02, $P = 0.779$ for % ±1.00 D by SRK/T-Kane); Haigis exhibited nonsignificant inferiority (Fig 4B, RR 0.99, $P = 0.859$ for % ±0.50 D by SRK/T-Haigis; RR 0.97, $P = 0.306$ for % ±1.00 D by SRK/T-Haigis), while the performance of BUII relative to SRK/T turned out to be contradictory for the predicted refractive error within different ranges, yet both being insignificant (RR 0.92, $P = 0.427$ for % ±0.50 D and RR 1.02, $P = 0.671$ for % ±1.00 D).

## Network meta-analysis

A network meta-analysis was further conducted concerning the accuracy of IOL power calculation formulas in PAC/PACG eyes based on the studies included. Pairwise comparisons were realized according to both direct and indirect evidence, with the results exhibited as forest plots in Fig 5 using the widely used SRK/T as a reference for all the MAE, % ±0.50 D and % ±1.00 D comparisons. 5 formulas were compared for MAE and 6 for % ±0.50 D and % ±1.00

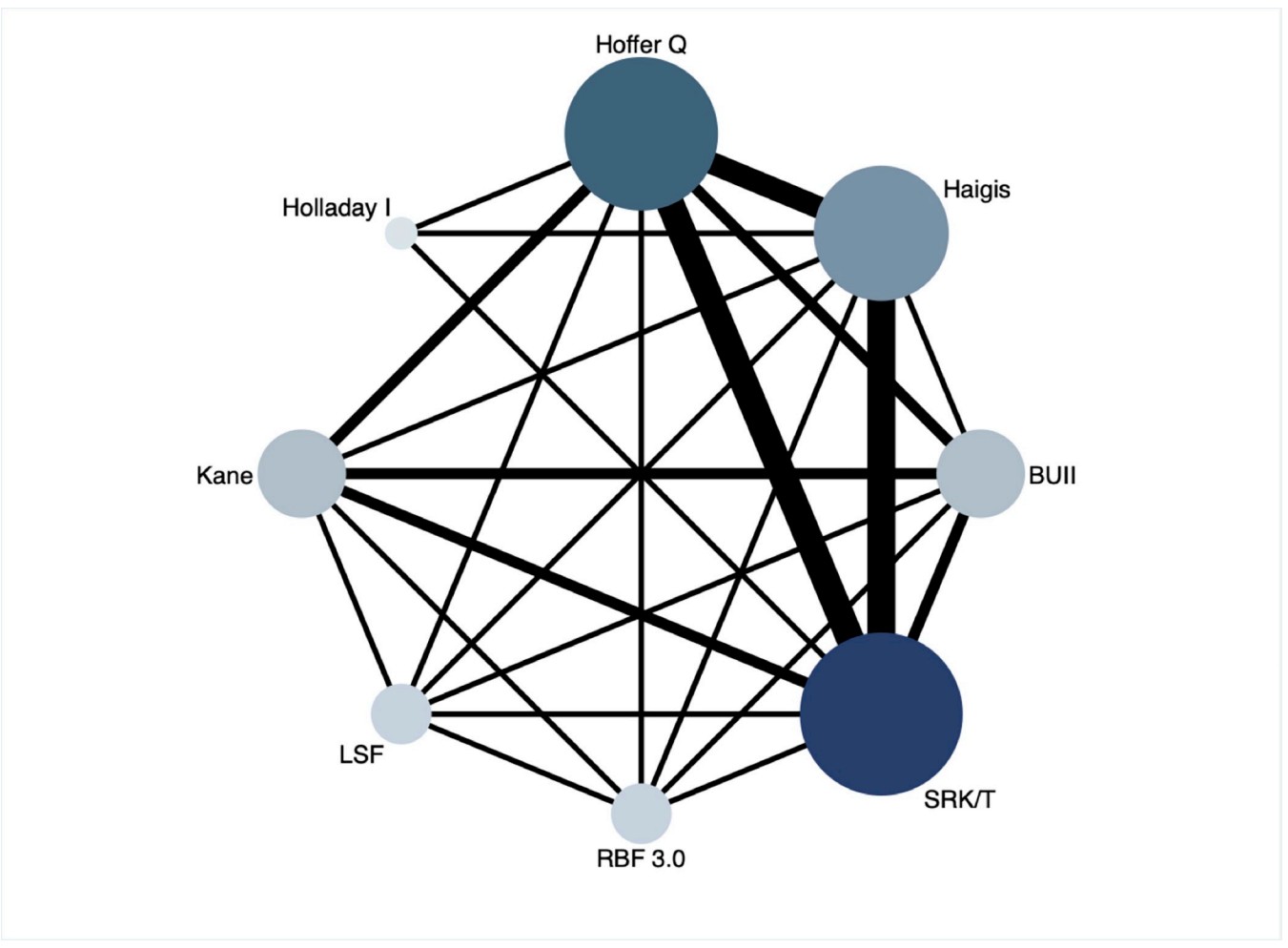

**Fig 2. Overall network map of 8 calculation formulas involved for comparison.** Each node represents 1 formula, with the size of the node proportional to the numbers of eyes enrolled using the corresponding formula. The edges that connect nodes suggest direct comparisons extracted from original studies, with the width proportionate to the number of studies. SRK/T: Sanders, Retzlaff, Kraff/theoretical; BUII: Barrett Universal II; RBF: Hill-Radial Basis Function; LSF: Ladas Super Formula.

D after data eligibility evaluation. All the studies passed the consistency test (S1 and S2 Figs), and the trace and density plot of network comparison indicated satisfactory convergence of the model. Risk of publication bias of MAE, % ±0.50 D and % ±1.00 D was measured using network funnel plot (S3A–S3C Fig for MAE, % ±0.50 D and % ±1.00 D, respectively), which appeared symmetrical and therefore indicated relatively small publication bias. Generally, there were no significant difference disclosed by any pairwise comparison, yet statistical analysis revealed potential superiority or inferiority. Specifically, BUII (MD -0.12, 95% CI -0.29 to 0.044) and Kane (MD -0.046, 95% CI -0.20 to 0.12) appeared insignificantly superior to SRK/T regarding MAE (Fig 5A). Additionally, pooled results showed both RBF 3.0, Kane and Hoffer Q with probably higher percentage of prediction error within % ±0.50 D, while all formulas except for Haigis seemed to beat SRK/T concerning % ±1.00 D (Fig 5B and 5C). In either MAE or % ±0.50/1.00 D, Haigis might underperform SRK/T, and LSF was likely to exhibit worse prediction accuracy in terms of % ±0.50 D.

To rank these formulas, we further calculated the rank possibilities of 8 formulas at each position based on their performance as measured by MAE, % ±0.50 D and % ±1.00 D. The

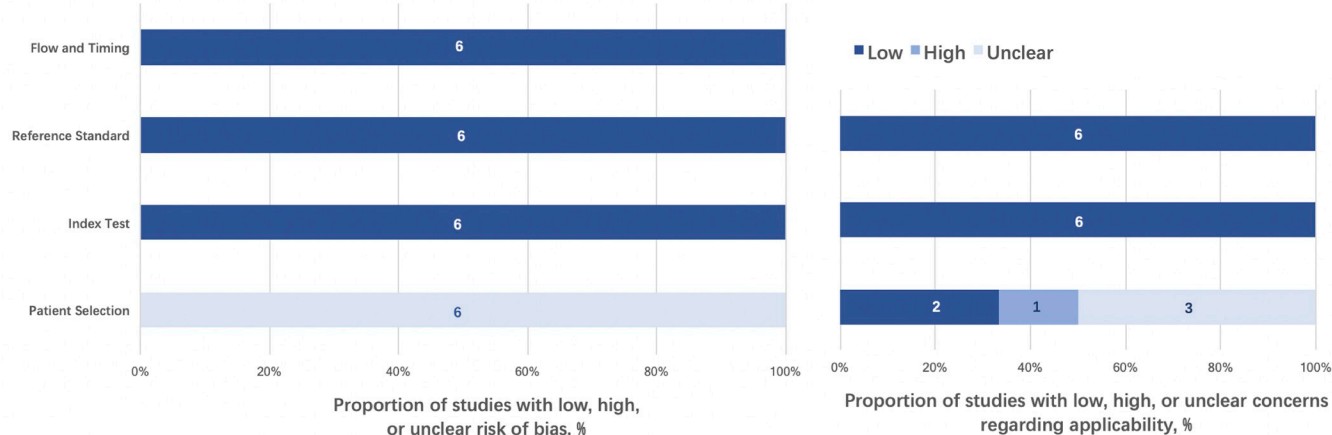

**Fig 3. Graphical display for quality assessment of selected studies using modified QUADAS-2 tool.** All of the studies were rated as 'unclear' in terms of the potential bias in patient selection, since they were all retrospective and did not report the enrollment strategy of patients (consecutively or randomly). As the exact meanings of 'diagnostic tests', 'index tests' or 'reference standard' differed from their original interpretations in this research field, the risk for bringing 'index tests' bias was considered low after our careful evaluation. Since all studies used SRK/T as a relative reference, the risk of bias was regarded as low for 'reference standard'. Regarding time flow, all studies were of high quality. QUADAS-2: Quality Assessment of Diagnostic Accuracy Studies-2.

weighed overall ranks of 6 formulas according to the MAE and of 7 formulas according to % ±0.50 D and % ±1.00 D were plotted in Fig 6A–6C, respectively, showing the potential efficacy of these methods in predicting IOL power in patients with PAC and PACG. Specifically, BUII outperformed in controlling MAE, followed by Kane (71.1% and 14.7% probabilities for being 1st, respectively). However, the ranks of % ±0.50 D and % ±1.00 D showed variability. For % ±0.50 D, the RBF 3.0, Kane and Hoffer Q formulas showed 77.4%, 80.1% and 56.1% of the possibilities to rank the top 3, respectively; while in terms of % ±1.00 D, the possibilities were 69.0% for Hoffer Q, 53.9% for both BUII and LSF for being top 3. For further confirmation of study consistency, node-splitting analysis was conducted to identify discrepancies between the direct and indirect effect of comparisons (S4 Fig), and the result showed that direct and indirect evidence were in agreement with each other and with the results of the consistency model.

## Discussion

Accuracy of IOL power calculation formulas applied in PACD patients has long been controversial. Nowadays, more than ten formulas have been developed and undoubtedly there are more on the way, into which various parameters are incorporated to improve calculation accuracy. Every formula is reasonably suggested to be used with respect to its specific range of application [27]. However, the fact that almost all the formulas were derived from normal population resulted in their application in PACD tantamount to external validation based on ill populations. Consequently, discrepancies are unsurprising in IOL prediction in eyes with PACD. Additionally, currently available studies conducted among PACD eyes showed divergent accuracies of different formulas [7, 9, 21, 24–26]. All these facts stress the importance of our study.

### Principal findings

In this network meta-analysis, we had undergone thorough literature search, enrolling 6 qualified studies with 419 eyes. Though with the intention of targeting PACD patients, our preliminary search revealed that all studies that were eligible mainly focused on conditions of

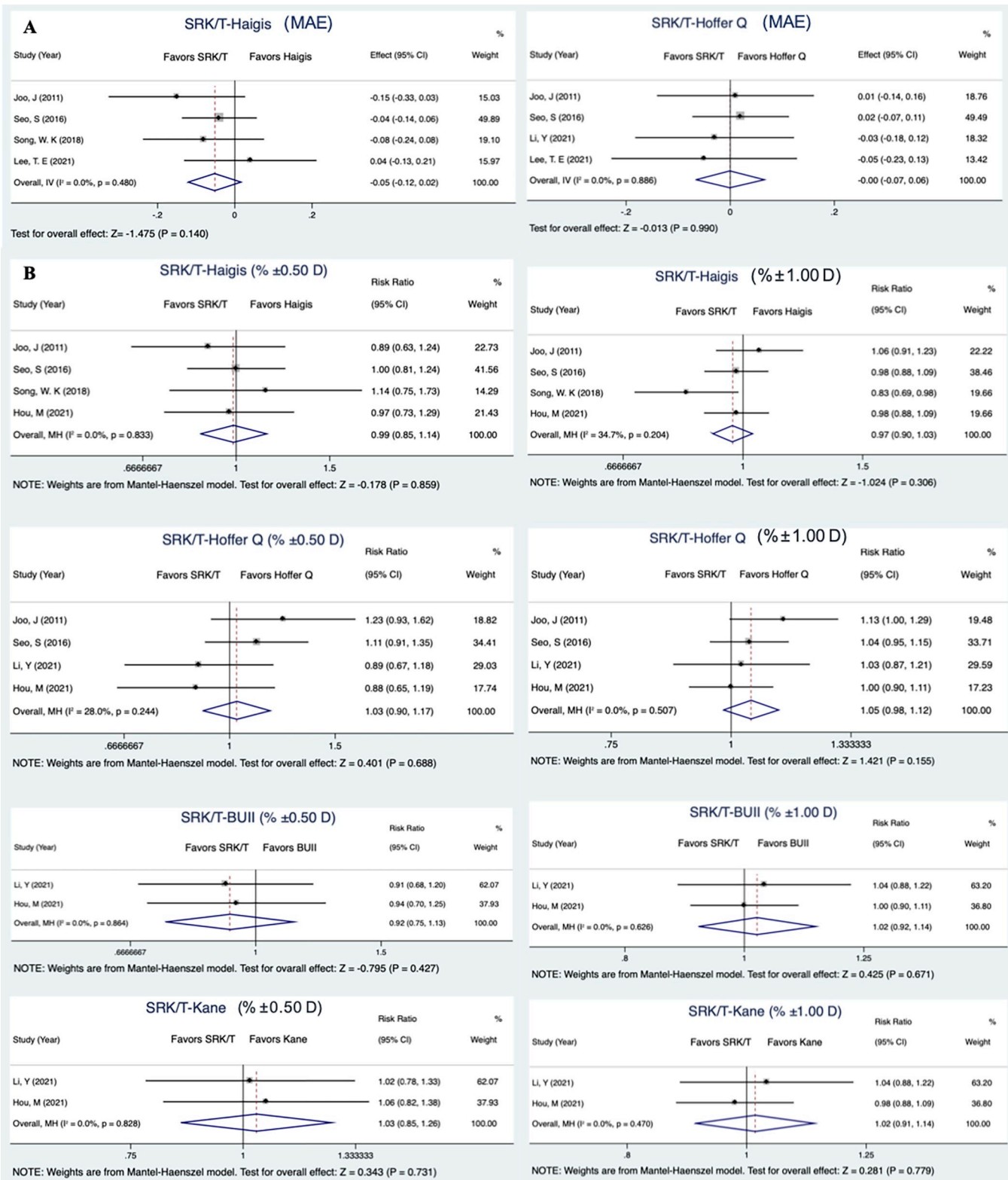

**Fig 4. The forest plots of traditional meta-analysis for the analysis of MAE and % ±0.50/1.00 D.** Direct comparisons of MAE between SRK/T and Haigis/ Hoffer Q (A) and % ±0.50/1.00 D between SRK/T and Haigis/Hoffer Q/BUII/Kane (B) were displayed in the forest plots. *P* value of Cochran's Q test (for MAE) or Mantel-Haenszel Q test (for % ±0.50/1.00 D) were reported for testing study heterogeneity. SRK/T: Sanders, Retzlaff, Kraff/theoretical; BUII: Barrett Universal II.

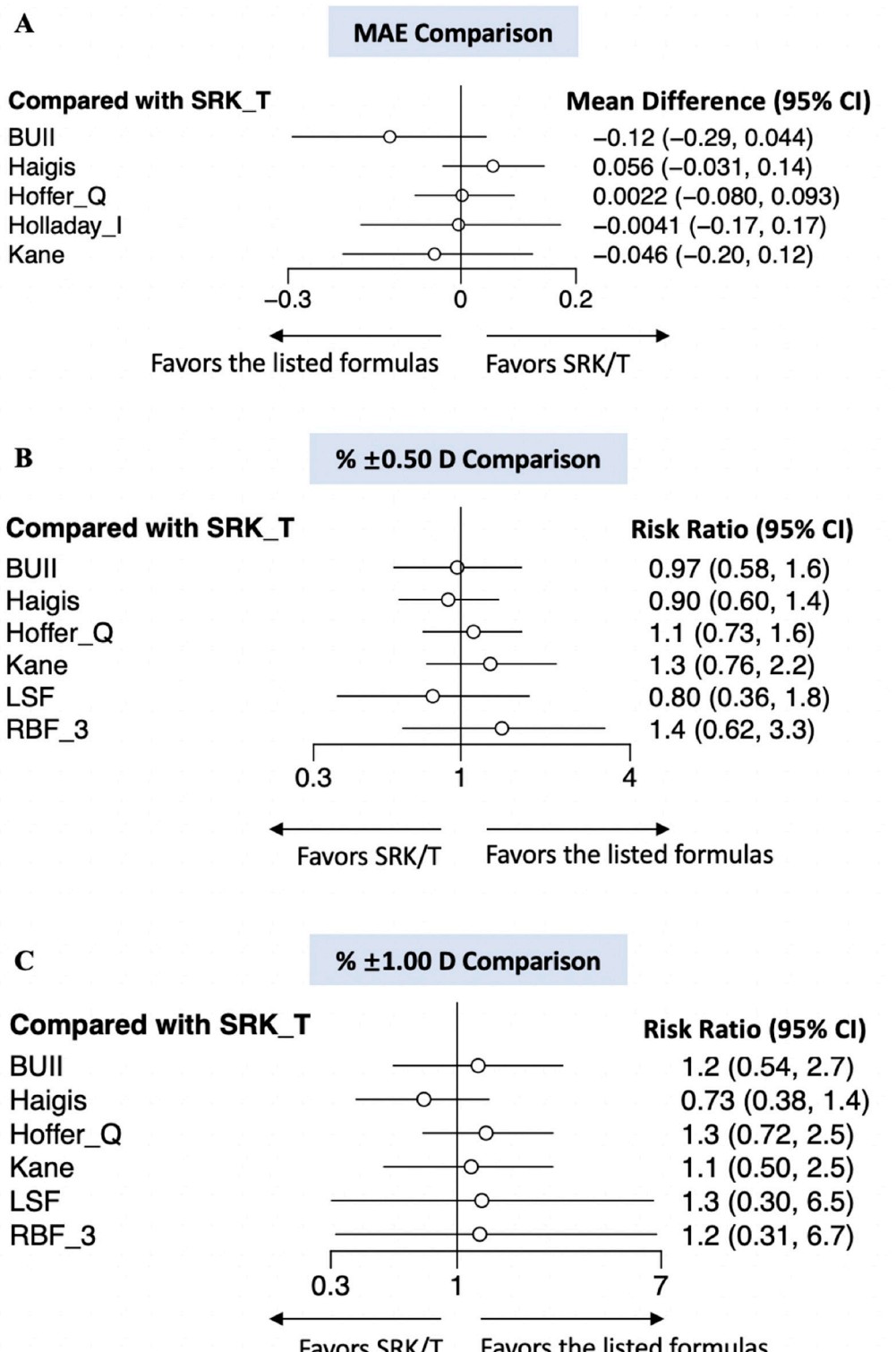

**Fig 5. The forest plots of the pairwise comparison for MAE and % ±0.50/1.00 D.** The SRK/T formula was set as a reference and was compared with other listed methods. The pooled results for comparison of MAE, % ±0.50 D and % ±1.00 D were illustrated in A, B and C, respectively. SRK/T: Sanders, Retzlaff, Kraff/theoretical; BUII: Barrett Universal II; RBF: Hill-Radial Basis Function; LSF: Ladas Super Formula.

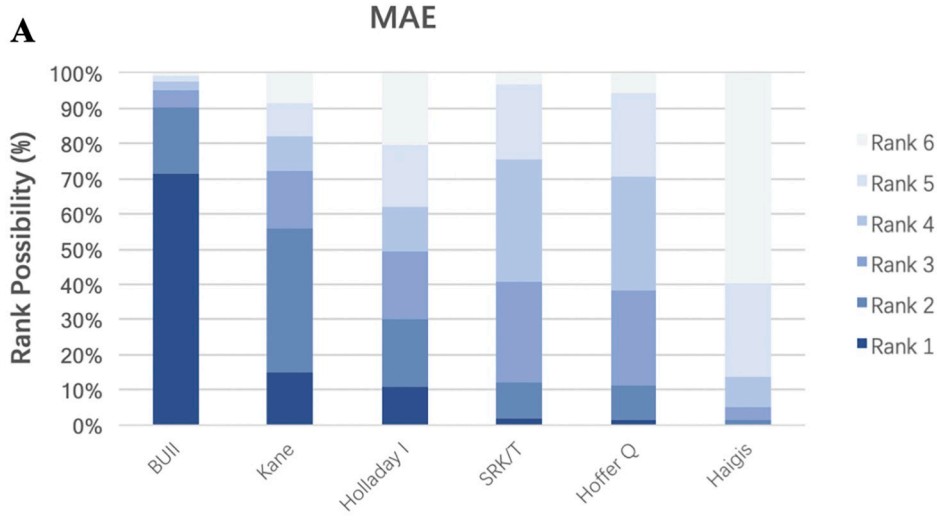

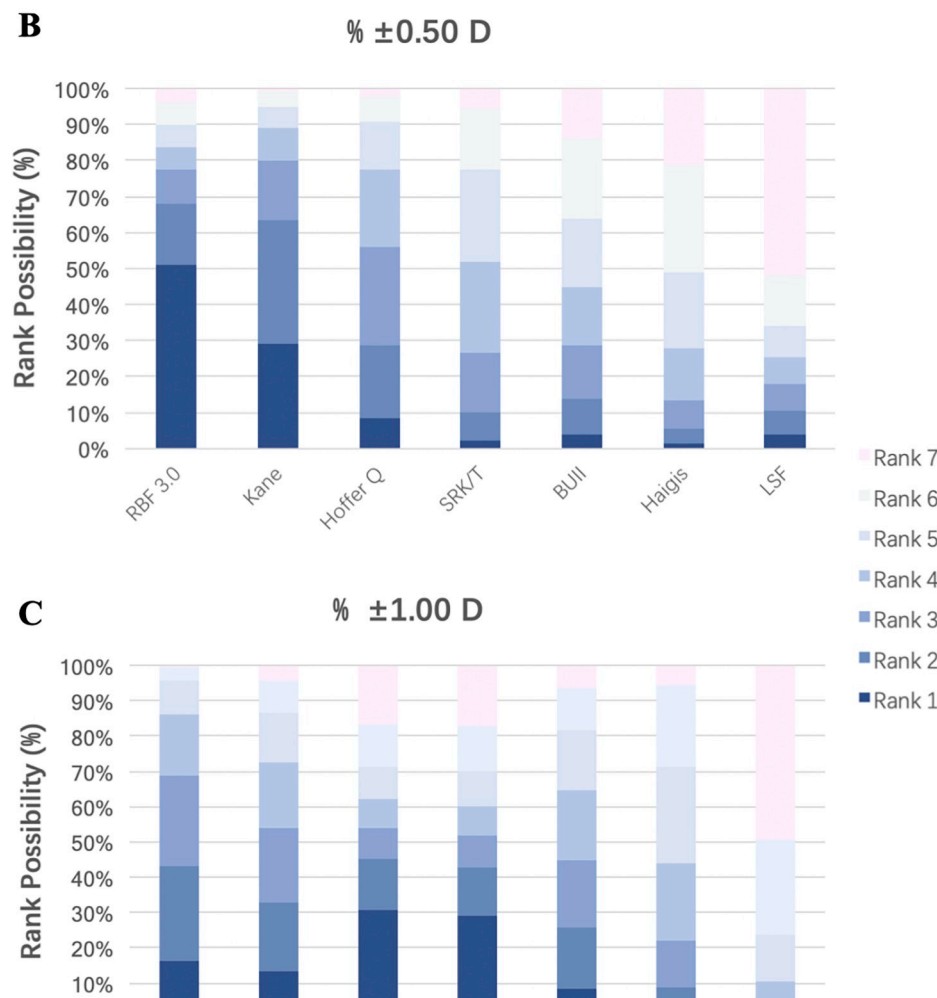

**Fig 6. Ranking probabilities (%) of formulas according to their performance on MAE, % ±0.50 D and % ±1.00 D.**
Ranking possibility showed the probabilities of each formula at different positions when considering their performance
of prediction accuracy, which was assessed by MAE (A) and prediction error (% ±0.50 D (B) and % ±1.00 D (C)). All
the formulas were plotted in descending order. SRK/T: Sanders, Retzlaff, Kraff/theoretical; BUII: Barrett Universal II;
RBF: Hill-Radial Basis Function; LSF: Ladas Super Formula.

synechial angle closure without/with glaucomatous neuropathy in the PACD spectrum (i.e.,
PAC, PACG). Though PACS was also mentioned in some studies [9], the fact that it did not
strictly follow the ISGEO definition led to our exclusion of the data in case any bias were intro-
duced. Furthermore, given that less than 1% of PACS would progress into PAC/PACG per
year as stated by He et al. [28] and that only 2.5~3.75% PACS ultimately develop PAC/PACG
[29], PACS might be an independent subgroup different from PAC/PACG in view of patho-
genesis and prognosis. Consequently, we only included studies focusing on PAC and PACG
for further analysis. QUADAS-2 were applied (Fig 3) with further identification of risk of bias
of included studies (S2 Table), and in general, all the studies included showed acceptable risk
of bias.

Somewhat pitifully, our results of both traditional and network meta-analysis showed no
significant differences among all the included formulas concerning the refractive errors being
introduced. However, there were signs that some formulas might outperform the others. For
traditional meta-analysis, direct comparisons of eligible formulas with SRK/T indicated that
Kane, a new-generation formula [30], showed slight but nonsignificant superiority over SRK/
T for % ±0.50 D and % ±1.00 D. The outperformance of new-generation formulas could be
attributed to the incorporation of various essential ocular parameters involved in pathogenesis
of primary angle-closure conditions [30, 31], yet more were to be discovered of the exact rea-
sons behind the potential error. The contradictory performance of BUII, one of the most accu-
rate and widely-used new-generation formulas, could be explained by its mere evaluation in
only 2 enrolled studies, for which bias might exist. We were also surprised by the mismatch
between clinical practice and our findings, i.e., the underperformance of Haigis and the out-
performance of rarely-used Hoffer Q. But all these conclusions should be drawn with care as
none of the comparison appeared statistically significant. For network meta-analysis, BUII
achieved better postoperative outcomes than other formulae based on MAE, consistent with
previous studies and current clinical experience [8]. However, BUII and some other seemingly
preferable formulas like RBF 3.0 and LSF had relatively wide confidence intervals, indicating
their instability when applied for PAC/PACG eyes. Altogether with the analysis of ranking
possibilities, IOL calculation formulas varied in their performance, thereby further demon-
strating the nonexistence of their absolute advantages over each other in PAC/PACG eyes.

## Limitations of this study

The major limitation of this study was the limited studies eligible for this meta-analysis, which
on the one hand hindered the revealing of clinically significant findings and on the other hand
made further subgroup analyses less possible. As for the reasons, firstly, inadequate clinical
researches were conducted thus far in this field, owing to the easily-misinterpreted dissatisfac-
tory postoperative visual quality for PACD patients as a result of glaucomatous neuropathy
rather than caused by predicted refractive error of implanted IOL. Moreover, we found a
greater number of studies focusing on the assessment of IOL power calculation formulas with
respect to short AL or shallow AC [13, 30, 32–34], which were seemingly associated with
angle-closure conditions. Nevertheless, short AL or shallow AC were not necessarily the major
or only factors that contribute to PACD. Any abnormality in iris configurations, lens thick-
ness, ciliary body conformation, zonular fiber tension or choroidal volume could contribute to

the occurrence of PACD [1]. In a word, PACD, especially PAC/PACG, is a spectrum of diseases with complex pathogenesis. Hence, we chose to conduct this meta-analysis of formula accuracy on PACD patients instead of on those with solely shorter AL or shallow AC.

In terms of the outcome measurements being evaluated and statistically analyzed, MAE was not necessarily the best choice. As a continuous data of averaged absolute error (AE) brought by different formulas, MAE were to be easily influenced by the distribution pattern of the original data. Since AE usually does not conform to Gaussian distribution, the MAE with its standard deviation is probably not the appropriate measure of central tendency and variability of the data. Nevertheless, as MAE had been introduced as a major outcome measurement by almost all the studies, it was still analyzed in this meta-analysis for a preliminary look at whether it was consistent with the percentages of cases within the refractive limits. For cases of asymmetric distributions of outcome measurements, the median absolute error (MedAE) would be a better representative of the central tendency, yet only a few studies provided the MedAE by formulas with even less information on the range of the original data [7, 24, 26], resulting in between-study comparisons of MedAE infeasible. Additionally, Holladay et al. recommended that SD of prediction error (PE) has shown to be perfect and tightly correlated with other parameters of assessing the accuracy of a IOL power calculation formula [35], however as SD is a random variable, it is not applicable in network meta-analysis. Furthermore, % ±0.50 and % ±1.00 D, as being available in most studies, were also recommended for revealing central and dispersion tendency regardless of the data distribution, which was thereby considered as being more reliable when evaluating formula accuracy.

## Excluded studies

As a systemic review and network meta-analysis, some relevant studies carried out in PACD populations were excluded for quantitative analysis for certain considerations. One of the situations was two studies with PACD patients implanted with 3-piece IOL, one showing unexpected and obvious discrepancies with 1-piece IOL-implantation [36], and the other providing unavailable outcome variables [11]. Though in clinical practice nowadays, not a few specialists hold that 3-piece IOLs would result in better clinical outcomes in terms of deepening and stabilizing ACD, the limited researches made it impossible for us to draw a conclusion as to which type of IOL outperforms with regard to postoperative refraction when applied in PACD patients. Meanwhile, the unclear degree of discrepancies brought by the IOL type led to exclusion of this study. Further evaluation and subgroup analysis could be done in the future to demonstrate different behaviors between one-piece and three-piece IOL in the capsular bag only if more researches were to be conducted. Another scenario was that some studies enrolled PACD patients with prior anti-glaucoma surgeries other than LPI (mainly trabeculectomy). Given it had already been demonstrated that cataract surgery in patients with prior trabeculectomy had significantly greater refractive surprise than those in the control groups [19, 20], these studies were also excluded. We had also ruled out patients with ongoing acute angle-closure episodes in case that corneal edema and intraocular inflammation would pose unexpected effects on IOL power prediction, though those with signs of previous acute attack were deemed as acceptable after further evaluation.

## Implications for future clinical practice and research

The clinical significance of this systemic review and meta-analysis was firstly to identify the potential preferable IOL power calculation formulas with higher accuracy in PACD eyes; and secondly to give advice on future studies in this field. For potential preferable formulas, we suggested that BUII, Kane and RBF 3.0 might exert better clinical outcomes, yet the

modification of existing formulas or development of new formulas specifically for PACD eyes were the real way to get to the root of this unmet clinical need. To promote high-quality researches in the future, here are a few recommendations: 1) Researchers should mind that 1-piece and 3-piece IOL might exert distinct clinical outcomes, and dividing them apart when conducting analysis could be beneficial for evaluating both the formula accuracy and IOL stability in angle-closure eyes; 2) It is necessary to only include patients with no prior anti-glaucoma surgeries other than LPI, for no one could be sure about the exact degree of potential zonular fiber injuries, capsular disposition and irregular astigmatism resulted from prior surgical interventions; 3) MedAE, SD of PE and the ratio of cases within certain limits of AE are considered as being more preferable, for they are more reliable and rational than MAE. Besides, as Holladay et al. suggested, attention should be paid if unequal variance exists among outcome variables derived from different formulas, where heteroscedastic method for data analysis should be performed [35]; 4) For measurement accuracy, unified application of new-generation optical biometry like IOLMaster or Lenstar have been recommended for ocular biometry, and subjective manifest refraction is recommended for refractive measurements; 5) IOL-specific constants ought to be optimized according to Langenbucher et al. [37, 38], and a cross-validation strategy is advised if possible. If not, downloading constants from the User Group for Laser Interference Biometry (ULIB) or using the Excel Query/What IF function [7] are other options; 6) Eyes with a postoperative visual acuity less than 0.4 should not be enrolled for formula constant optimization or accuracy assessment. All these aspects send expectations for the future development of both high-quality studies and more appropriate formulas for PACD patients, the breakthrough of which lying in the identification of essential factors that significantly lead to postoperative refractive errors in PACD patients, e.g., the lens vault (LV) [7], the ACD/AL, the relative position of ciliary process, etc. On the basis of these factors, new formulas might be generated, bringing hope and brighter future for those who expect to be rescued from the otherwise possibly dissatisfactory visual outcomes.

## Conclusion

This systemic review and network meta-analysis of limited evidence suggested that no significant superiority was revealed among the commonly used SRK/T, Hoffer Q, Holladay 1, Haigis, BUII, Kane, RBF 3.0 and LSF formulas in eyes with PAC/PACG. Further well-designed studies are warranted to validate the efficacy of various formulas among PACD, yet modified or even new formulas derived specifically for PACD patients should come into being if none of these current formulas are stably satisfactory.

## Supporting information

**S1 Checklist. Adoption of the PRISMA 2009 checklist for reporting of systematic reviews and meta-analyses.**
(DOC)

**S1 Fig. Consistency test for MAE.** Between-study heterogeneity of MAE was measured using mtc.anohe package in R, with the results showing consistency in both pair-wise and network comparison.
(TIFF)

**S2 Fig. Consistency test for % ±0.50 D and % ±1.00 D.** Between-study heterogeneity of A. % ±0.50 D and B. % ±1.00 D was measured using mtc.anohe package in R, with the results showing consistency in all the network comparison.
(TIFF)

**S3 Fig. Network funnel plot.** Risk of publication bias of A. MAE, B. % ±0.50 D and C. % ±1.00 D was measured using network funnel plot showing symmetry, indicating relatively small publication bias.
(TIFF)

**S4 Fig. The plotted node-splitting analysis.** The decision rule selected Haigis-Hoffer Q comparisons for MAE (A), Haigis-BUII, Haigis-Hoffer Q and Haigis-Kane comparisons for % ±0.50 D (B) and % ±1.00 D (C) because of presence of both direct and indirect comparisons in these pairs. As reflected by P-value, the direct and indirect comparisons were in agreement of each other. MAE: mean absolute error; BUII: Barrett Universal II.
(TIFF)

**S1 Table. Detailed search results from different sources.**
(DOCX)

**S2 Table. Risk of bias of included studies.** Level of evidence as recommended by the Oxford Centre for Evidence-based Medicine. *Retrospective Observational Studies.
(PDF)

## Author Contributions

**Conceptualization:** Hongfang Yang, Xinghuai Sun.

**Data curation:** Wenhan Lu.

**Formal analysis:** Wenhan Lu, Yu Hou.

**Funding acquisition:** Hongfang Yang, Xinghuai Sun.

**Methodology:** Wenhan Lu.

**Supervision:** Hongfang Yang, Xinghuai Sun.

**Writing – original draft:** Wenhan Lu.

**Writing – review & editing:** Wenhan Lu, Yu Hou, Hongfang Yang, Xinghuai Sun.

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
