## [Decision Letter · Decision Letter 0]

19 Sep 2022

PONE-D-22-24306A Systemic Review and Network Meta-analysis of Accuracy of Intraocular Lens Power Calculation Formulas in Primary Angle-closure ConditionsPLOS ONE

Dear Dr. Lu,

Thank you for submitting your manuscript to PLOS ONE. After careful consideration, we feel that it has merit but does not fully meet PLOS ONE’s publication criteria as it currently stands. Therefore, we invite you to submit a revised version of the manuscript that addresses the points raised during the review process.

We look forward to receiving your revised manuscript.

Kind regards,

Andrzej Grzybowski

Academic Editor

PLOS ONE

Journal Requirements:

Reviewers' comments:

Reviewer's Responses to Questions

**Comments to the Author**

1. Is the manuscript technically sound, and do the data support the conclusions?

Reviewer #1: Yes

Reviewer #2: No

2. Has the statistical analysis been performed appropriately and rigorously? 

Reviewer #1: Yes

Reviewer #2: No

3. Have the authors made all data underlying the findings in their manuscript fully available?

Reviewer #1: Yes

Reviewer #2: Yes

4. Is the manuscript presented in an intelligible fashion and written in standard English?

Reviewer #1: Yes

Reviewer #2: Yes

5. Review Comments to the Author

Reviewer #1: The authors conducted a comprehensive literature search to assess the success of IOL power calculation with 8 formulas in open angle glaucoma eyes. The conclusion was that there was no absolute advantage for PAC/PACG eyes among the formulas included in this study.

Abstract, background: „the refractive error after intraocular lens implantation always deviates from anticipation.” The wording is strange, and the use of „always” is a bit strong.

Abstract, methods: „…within 0.50 diopiters (D) or 1.00 D of estimated refraction”. This is more like „prediction error”.

It was very interesting to read that only two studies used BUII and/or the most modern methods using artificial intelligence. In addition, the known problem of MAE vs. MedAE is also well described and makes it difficult to draw a conclusion that can be used in everyday practice.

This systemic review is worthy of publication after correcting the above-mentioned minor issues.

Reviewer #2: The authors have put a tremendous amount of effort and thought in the study design. Unfortunately, a major limitation is the wide variation in the formulas used in the various studies, making it difficult to draw meaningful conclusions beyond those that are available in recent studies that include the more modern formulas.

Line 106: The “S” of PACS was not stated. Presumably it refers to “suspect.”

Why was the SRK-T chosen as the reference formula to compare with all others? Also, the authors do not indicate that SRK-T is used in this way in the abstract.

I did not see a test for normality of the data, and I strongly suspect that the data are not Gaussian. Therefore, the heteroscedastic formulae are recommended as described by Holladay et al (Holladay JT, Wilcox RR, Koch DD, Wang L. J Cataract Refract Surg 2021 Jan;47 (1):65-77) with the formulas in R, using the McNemar for analyzing % values (oph.astig.mcnemar) and the oph.astig.depcom for comparing mean absolute errors.

In several places, the authors mention “nonsignificant” and “trend”—only differences that were significant are valid. Comments about trends should be deleted.

6. PLOS authors have the option to publish the peer review history of their article (what does this mean?). If published, this will include your full peer review and any attached files.

Reviewer #1: No

Reviewer #2: No

---

## [Author Response · Author response to Decision Letter 0]

25 Sep 2022

We would like to thank the editor and assigned reviewers for their careful reading of the manuscript and for the insightful and constructive comments and suggestions. In this revised manuscript, we have responded to each point of the reviewer. 

Reviewer #1: 

The authors conducted a comprehensive literature search to assess the success of IOL power calculation with 8 formulas in open angle glaucoma eyes. The conclusion was that there was no absolute advantage for PAC/PACG eyes among the formulas included in this study.

 1) Abstract, background: “the refractive error after intraocular lens implantation always deviates from anticipation.” The wording is strange, and the use of “always” is a bit strong. 

Answer: We thank the reviewer for pointing out the improper wording. We have modified the language in the corresponding positions (line 24-27).

 2) Abstract, methods: “…within 0.50 diopiters (D) or 1.00 D of estimated refraction”. This is more like “prediction error”.

Answer: We have modified this sentence into “Primary outcomes were the mean absolute errors (MAE) and the percentages of eyes with a prediction error within 0.50 diopiters (D) or 1.00 D (% 0.50/1.00 D) by different formulas” in line 32-34.  

It was very interesting to read that only two studies used BUII and/or the most modern methods using artificial intelligence. In addition, the known problem of MAE vs. MedAE is also well described and makes it difficult to draw a conclusion that can be used in everyday practice.  This systemic review is worthy of publication after correcting the above-mentioned minor issues.

Answer: We thank the reviewer for holding positive attitudes towards our study.

Reviewer #2: 

The authors have put a tremendous amount of effort and thought in the study design. Unfortunately, a major limitation is the wide variation in the formulas used in the various studies, making it difficult to draw meaningful conclusions beyond those that are available in recent studies that include the more modern formulas. Answer: We also felt pity for not being able to draw meaningful conclusions beyond those that are available in recent studies, and that’s why we urge for more studies to be conducted. 

 1) Line 106: The “S” of PACS was not stated. Presumably it refers to “suspect.”

Answer: We thank the reviewer for pointing out the missing of statement. We have added the full name for PACS (line 106-110).  

2) Why was the SRK-T chosen as the reference formula to compare with all others? Also, the authors do not indicate that SRK-T is used in this way in the abstract.

Answer: For the traditional meta-analysis, head-to-head comparisons were conducted between SRK/T formula and the others since the SRK/T was the most popular IOL power calculating formula applied in angle-closure patients as being investigated in all the studies included. We added more explanation in methods part in the abstract (line 37), and have also mentioned that in line 192-193 (“SRK/T had been investigated in all the studies included, therefore showing strongest links with the other formulas”), line 215-216 (“Since all studies used SRK/T as a relative reference, the risk of bias was regarded as low for ‘reference standard’”), and line 220-222 (“…between the most commonly used SRK/T formula (which was evaluated by all the studies included) and the others”), with the hope of clarifying this point.  

3) I did not see a test for normality of the data, and I strongly suspect that the data are not Gaussian. Therefore, the heteroscedastic formulae are recommended as described by Holladay et al (Holladay JT, Wilcox RR, Koch DD, Wang L. J Cataract Refract Surg 2021 Jan;47 (1):65-77) with the formulas in R, using the McNemar for analyzing % values (oph.astig.mcnemar) and the oph.astig.depcom for comparing mean absolute errors.

Answer: We thank the reviewer for providing insightful opinions on statistical analysis. Here, we would like to respond from the following perspectives:

1. For the outcome measurements we used, MAE, as a continuous data of averaged absolute error brought by different formulas, were to be easily influenced by the distribution pattern of the original data. % 0.50 and % 1.00 D, however, as binary variable, were less likely to be influenced, so we mainly focus on MAE when discussing the test for normality. As a meta-analysis, we were unable to directly analysis data normality for the lack of original data, and data normality could only be inferred if the authors reported. Among all the studies being included, Li et al had verified the normality of distribution using the Shapiro-Wilk normality test; Lee et al had reported none-Gaussian distribution and therefore used the Mann–Whitney U-test to compare the differences. Other studies did not mention the normality of the data. We suggested, as stated in our discussion part that MAE was not the best choice for data analysis. However, for clarification, we stated why we still chose MAE as one of the outcome variables (line 365-368);

2. Consistency, similarity and transitivity are strict methodological assumptions that a network meta-analysis requires[1], based on which we had carefully tested the data reliability among all the included studies. We have conducted consistency test using mtc.anohe package in R and node-splitting test in our work, with all the results showing reliable (S1-S2 and S4 Fig).

3. We have, as the reviewer recommended, carefully read the heteroscedastic formulae recommended as described by Holladay et al. And as stated in the literature, standard deviation (SD) of prediction error (PE) has shown to be perfect and tightly correlated with other parameters of assessing the accuracy of a IOL power calculation formula. We have intended to conduct further analysis on SD of PE, however as SD is a random variable, it is not applicable in network meta-analysis. In that case, MAE, % 0.50 D (or % 1.00 D) were still used as outcome variables. But we still put additional explanation with regard to this point into our manuscript (line 372-375, line 414-416)

4. For future studies, we included in our work the urge for researchers to consider more carefully about choosing proper outcome variables and means for data analysis (line 416-419), therefore providing more robust and reliable evidence on IOL power calculation formulae selection. 

 In several places, the authors mention “nonsignificant” and “trend”—only differences that were significant are valid. Comments about trends should be deleted.

Answer: We do agree that some improper wording with regard to the “trend” had probably be misleading. We have made revision (line 234-237, line 257-266), and have also mentioned about this “unreliable trend” in the discussion part, hoping not to bring any ambiguity. 

We thank the editor and assigned reviewers again for their careful reading of the manuscript and for the insightful and constructive comments and suggestions. Hope all the works we have done would make our systemic review and meta-analysis readable and useful, and please do not hesitate to contact us if there is any question.

References:

1. Shim S, Yoon BH, Shin IS, Bae JM. Network meta-analysis: application and practice using Stata. Epidemiol Health. 2017;39:e2017047. Epub 2017/11/03. doi: 10.4178/epih.e2017047. PubMed PMID: 29092392; PubMed Central PMCID: PMCPMC5733388.

---

## [Decision Letter · Decision Letter 1]

4 Oct 2022

A Systemic Review and Network Meta-analysis of Accuracy of Intraocular Lens Power Calculation Formulas in Primary Angle-closure Conditions

PONE-D-22-24306R1

Dear Dr. Lu,

We’re pleased to inform you that your manuscript has been judged scientifically suitable for publication and will be formally accepted for publication once it meets all outstanding technical requirements.

Kind regards,

Andrzej Grzybowski

Academic Editor

PLOS ONE

Additional Editor Comments (optional):

Reviewers' comments:

Reviewer's Responses to Questions

**Comments to the Author**

1. If the authors have adequately addressed your comments raised in a previous round of review and you feel that this manuscript is now acceptable for publication, you may indicate that here to bypass the “Comments to the Author” section, enter your conflict of interest statement in the “Confidential to Editor” section, and submit your "Accept" recommendation.

Reviewer #1: All comments have been addressed

Reviewer #2: All comments have been addressed

2. Is the manuscript technically sound, and do the data support the conclusions?

Reviewer #1: Yes

Reviewer #2: Yes

3. Has the statistical analysis been performed appropriately and rigorously? 

Reviewer #1: Yes

Reviewer #2: Yes

4. Have the authors made all data underlying the findings in their manuscript fully available?

Reviewer #1: Yes

Reviewer #2: Yes

5. Is the manuscript presented in an intelligible fashion and written in standard English?

Reviewer #1: Yes

Reviewer #2: Yes

6. Review Comments to the Author

Reviewer #1: The revised manuscript is OK in its current form. All questions are answered properly, no additional problems to fix.

Reviewer #2: Thanks for your detailed and thoughtful responses. I appreciate the effort you made to address the issue of normality of the data.

7. PLOS authors have the option to publish the peer review history of their article (what does this mean?). If published, this will include your full peer review and any attached files.

Reviewer #1: No

Reviewer #2: No

---

## [Editor Report · Acceptance letter]

6 Oct 2022

PONE-D-22-24306R1 

A Systemic Review and Network Meta-analysis of Accuracy of Intraocular Lens Power Calculation Formulas in Primary Angle-closure Conditions 

Dear Dr. Lu:

I'm pleased to inform you that your manuscript has been deemed suitable for publication in PLOS ONE. Congratulations! Your manuscript is now with our production department. 

Kind regards, 

on behalf of

Dr. Andrzej Grzybowski 

Academic Editor

PLOS ONE